# High performance magnesium-based plastic semiconductors for flexible thermoelectrics

Airan Li [1,4], Yuechu Wang [1,4], Yuzheng Li[1,2], Xinlei Yang[1], Pengfei Nan[3], Kai Liu[1], Binghui Ge [3], Chenguang Fu [1] ✉ & Tiejun Zhu [1,2] ✉

Low-cost thermoelectric materials with simultaneous high performance and superior plasticity at room temperature are urgently demanded due to the lack of ever-lasting power supply for flexible electronics. However, the inherent brittleness in conventional thermoelectric semiconductors and the inferior thermoelectric performance in plastic organics/inorganics severely limit such applications. Here, we report low-cost inorganic polycrystalline $Mg_3Sb_{0.5}Bi_{1.498}Te_{0.002}$, which demonstrates a remarkable combination of large strain (~ 43%) and high figure of merit $zT$ (~ 0.72) at room temperature, sur-passing both brittle $Bi_2(Te,Se)_3$ (strain ≤ 5%) and plastic $Ag_2(Te,Se,S)$ and organics ($zT ≤ 0.4$). By revealing the inherent high plasticity in $Mg_3Sb_2$ and $Mg_3Bi_2$, capable of sustaining over 30% compressive strain in polycrystalline form, and the remarkable deformability of single-crystalline $Mg_3Bi_2$ under bending, cutting, and twisting, we optimize the Bi contents in $Mg_3Sb_{2-x}Bi_x$ ($x = 0$ to 1) to simultaneously boost its room-temperature thermoelectric performance and plasticity. The exceptional plasticity of $Mg_3Sb_{2-x}Bi_x$ is further revealed to be brought by the presence of a dense dislocation network and the persistent Mg-Sb/Bi bonds during slipping. Leveraging its high plasticity and strength, polycrystalline $Mg_3Sb_{2-x}Bi_x$ can be easily processed into micro-scale dimensions. As a result, we successfully fabricate both in-plane and out-of-plane flexible $Mg_3Sb_{2-x}Bi_x$ thermoelectric modules, demonstrating promising power density. The inherent remarkable plasticity and high thermoelectric performance of $Mg_3Sb_{2-x}Bi_x$ hold the potential for significant advancements in flexible electronics and also inspire further exploration of plastic inorganic semiconductors.

Thermoelectric (TE) materials have gained significant attention for their ability to convert heat into electricity in a solid-state form[1]. With the rapid advancement of flexible electronics[2], there is a growing demand for TE materials exhibiting both high performance and plas-ticity at room temperature for reliable and sustainable power generation[3,4]. Despite the discovery and development of numerous high-performance TE materials in inorganic semiconductors over the past century, they are inherently brittle[5]. Unlike metallic bonds in metals or polymer chains in organics, the directional covalent bonds in inorganic semiconductors impede atomic layer slipping, leading to

[1]State Key Laboratory of Silicon and Advanced Semiconductor Materials, School of Materials Science and Engineering, Zhejiang University, 310058 Hangzhou, China. [2]Shanxi-Zheda Institute of Advanced Materials and Chemical Engineering, Taiyuan 030000, China. [3]Information Materials and Intelligent Sensing Laboratory of Anhui Province, Key Laboratory of Structure and Functional Regulation of Hybrid Materials of Ministry of Education, Institutes of Physical Science and Information Technology, Anhui University, Hefei 230601, China. [4]These authors contributed equally: Airan Li, Yuechu Wang. ✉e-mail: chenguang_fu@zju.edu.cn; zhutj@zju.edu.cn

sudden and unpredictable breakdown under external forces[6]. For instance, commercially available $Bi_2Te_3$-based compounds typically withstand less than 5% compressive strain before fracturing abruptly[7,8]. The undesirable brittleness imposes severe limitations on the available processing methods, yield rates, and overall service lifespan of the TE semiconductors, thereby hindering their broader application in various fields, including but not limited to flexible electronics.

In recent years, several inorganic semiconductors with remarkable plasticity have been uncovered, primarily within the chalcogenide systems[9–15]. Notably, ZnS demonstrates a remarkable 45% compressive strain in darkness[9], while $Ag_2S$ and its alloys can endure over 10% tensile strain[10,16–20]. Additionally, van der Waals crystals like InSe[11] and $SnSe_2$[21] can exhibit obvious flexibility, easily being bent and curved. Compared to organic semiconductors, plastic inorganic semiconductors typically possess superior carrier mobility[22,23], making them promising candidates in flexible touch panels[24], memristors[25,26], and TE generators[3,27]. However, among these plastic inorganic semiconductors, only $Ag_2$(Te, Se, S) and $SnSe_2$ have been reported to exhibit moderate TE performance at room temperature, with a figure of merit ($zT = S^2\sigma T/\kappa$, where $S$, $\sigma$, $\kappa$ and $T$ represent Seebeck coefficient, electrical conductivity, thermal conductivity, and absolute temperature, respectively) of ~0.4[3,8,21], which are notably higher than that of plastic organic TE materials, but fall significantly short of that in conventional TE compounds. Currently, the absence of materials exhibiting simultaneous high room temperature TE performance and plasticity imposes a significant barrier to the advancement of flexible TE technology.

Low-cost n-type $Mg_3Sb_{2-x}Bi_x$ has attracted significant interest since its discovery due to its impressive high TE performance[28–31]. Peak $zT$ values of 1.5–1.8 at mid-temperatures (~773 K) have been achieved in $Mg_3Sb_{2-x}Bi_x$ with $x = 0.5$–1.0. Apart from its exceptional TE performance, abundant raw materials, high fracture toughness, and good machinability make n-type $Mg_3Sb_{2-x}Bi_x$ highly competitive for future practical applications, particularly at room temperature[32,33]. Optimizing the Bi content plays a significant role in enhancing the room-temperature performance of $Mg_3Sb_{2-x}Bi_x$. The synergic effects of reduced bandgap, increased band curvature, strengthened phonon scattering, and enhanced grain size by Bi alloying make $Mg_3Sb_{2-x}Bi_x$ ($x = 1.4$–1.75) exhibit $zT$ values above 0.6 at room temperature[34,35], which approach to that of the state-of-the-art n-type $Bi_2$(Te, Se)$_3$ and are much superior to the plastic $Ag_2$(Te, Se, S) and organic TE materials[31,36–39].

The remarkably high $zT$ values of $Mg_3Sb_{2-x}Bi_x$ have also sparked significant development in its TE modules over the past decades[40–42]. However, these advancements have predominantly focused on rigid TE modules, neglecting the potential application of $Mg_3Sb_{2-x}Bi_x$ in flexible electronics. Given its notable TE performance and potential plasticity at room temperature[36], there exists a compelling opportunity to explore the suitability of $Mg_3Sb_{2-x}Bi_x$ for power generation in flexible electronics. Achieving simultaneous high TE performance and plasticity in $Mg_3Sb_{2-x}Bi_x$ remains a challenge.

In this work, we first focus on binary $Mg_3Sb_2$ and $Mg_3Bi_2$, uncovering an exceptionally high compressive strain of >30% in polycrystalline samples. Remarkably, single-crystalline $Mg_3Bi_2$ also exhibits excellent deformability when subjected to bending, cutting, and twisting. By optimizing the Bi contents to modulate both TE performance and plasticity, a large compressive strain of 43% and a high $zT$ of 0.72 at room temperature are achieved simultaneously in polycrystalline $Mg_3Sb_{0.5}Bi_{1.498}Te_{0.002}$. The observed dense dislocation network is attributed to facilitating atomic layer slipping in $Mg_3Sb_{2-x}Bi_x$, while the persistent Mg-Sb/Bi bonds during slipping are crucial for maintaining structural integrity. Additionally, we demonstrate the good machinability of $Mg_3Sb_{2-x}Bi_x$, which can be processed to unbroken small granules and thin TE legs via dicing and cutting. By assembling thin TE legs on flexible substrates, we have successfully fabricated both in-plane and out-of-plane flexible modules based on polycrystalline $Mg_3Sb_{0.5}Bi_{1.498}Te_{0.002}$, showcasing their potential in flexible electronics. The simultaneous high plasticity and high TE performance of $Mg_3Sb_{2-x}Bi_x$, as well as the demonstration of flexible $Mg_3Sb_{2-x}Bi_x$ TE modules, pave the way for its utilization in flexible electronics and will spur the development of high-performance plastic TE semiconductors.

## Results

### High TE performance and remarkable plasticity in $Mg_3Sb_{2-x}Bi_x$

$Mg_3Sb_{2-x}Bi_x$ represents a solid solution combining $Mg_3Sb_2$ and $Mg_3Bi_2$. As depicted in Fig. 1a, both polycrystalline $Mg_3Sb_2$ and $Mg_3Bi_2$ exhibit remarkable plastic deformation, achieving over 30% strain under uniaxial compression. The optical images inset in Fig. 1a visually demonstrate their high plasticity, where polycrystalline $Mg_3Sb_2$ can be compressed from 6 mm to 3.6 mm. It should be mentioned that unlike $Mg_3Sb_2$, which breaks suddenly with a noticeable drop in the stress, $Mg_3Bi_2$ can be compressed after the first noticeable drop in the stress. The full compressive data for $Mg_3Bi_2$ is shown in Supplementary Fig. 1, which indicates that $Mg_3Bi_2$ can be compressed to about 1.6 mm (suggesting a compressive strain of about 80%) after experiencing two noticeable drops in the stress. However, upon examining the optical image of $Mg_3Bi_2$ after compression, it can be seen that $Mg_3Bi_2$ bulk shatters into some small pieces. Thus, it is not convincing to take this high compressive strain of 80% as the true compressive performance of $Mg_3Bi_2$. In addition, polycrystalline $Mg_3Sb_2$ and $Mg_3Bi_2$ also exhibit decent tensile strains of about 7.4% and 12.7%, respectively (Supplementary Fig. 2), surpassing numerous inorganic semiconductors and ceramics and even better than plastic $Ag_2S$ (4.2% elongation)[10]. The fracture surface morphology of polycrystalline $Mg_3Sb_2$ and $Mg_3Bi_2$ after compression is displayed in Supplementary Fig. 3 and Fig. 1b, where the fracture surface of polycrystalline $Mg_3Bi_2$ exhibits a notable wavy-like characteristic, indicative of its better plasticity and deformability than polycrystalline $Mg_3Sb_2$. Moreover, it is found that single-crystalline $Mg_3Bi_2$ in Fig. 1c also demonstrates remarkable deformability, which can be manually bent, twisted and cut, showcasing its potential for flexible electronics, while single-crystalline $Mg_3Sb_2$ cannot be bent and twisted like single-crystalline $Mg_3Bi_2$.

While both $Mg_3Sb_2$ and $Mg_3Bi_2$ exhibit favorable plasticity, their TE performances, particularly at room temperature, are rather inferior. Bi alloying is rather crucial in optimizing the room temperature $zT$ of $Mg_3Sb_2$[30,34]. As shown in Supplementary Fig. 4, Bi alloying has significant impacts on electrical transport properties of $Mg_3Sb_{2-x}Bi_x$. The downward shift of the peak $S$ in $Mg_3Sb_{2-x}Bi_x$ with higher Bi contents suggests that the bandgap is reduced, in consistency with previous reports[30–32,34]. Additionally, when Bi content ($x$) increases to 1.5, there is an obvious rise in room temperature $\sigma$, which suggests that the grain boundary scattering is weakened due to the larger grain sizes[30,34] (fracture morphology of $Mg_3Sb_{0.5}Bi_{1.5}$ in Supplementary Fig. 3). Moreover, Bi alloying leads to lower $\kappa$ due to the enhanced point defect scattering of phonons. As a result, for polycrystalline $Mg_3Sb_{2-x}Bi_x$, substantial improvement in room-temperature $zT$ is obtained after increasing the amounts of Bi ($x = 1.5$). A peak $zT$ of 0.72 has been attained at room temperature, while $zT$ values of 0.72–0.86 can be obtained within the near room temperature range (300–478 K) in $Mg_3Sb_{0.5}Bi_{1.498}Te_{0.002}$, rivaling the commercial high-cost polycrystalline $Bi_2$(Te,Se)$_3$. Besides the high TE performance achieved in $Mg_3Sb_{2-x}Bi_x$, as shown in Fig. 1e, all polycrystalline $Mg_3Sb_{2-x}Bi_x$ samples demonstrate considerable plastic deformation, with compressive strains exceeding 30%, displaying cracks upon finally fracturing. It can also be noticed that the fracture morphology of polycrystalline Bi-rich $Mg_3Sb_{2-x}Bi_x$ ($x = 1.5$) closely resembles that of $Mg_3Bi_2$ (Supplementary Fig. 3). Notably, polycrystalline $Mg_3Sb_{0.5}Bi_{1.498}Te_{0.002}$ achieves an impressive strain of 43%, overriding the Sb-rich $Mg_3Sb_{2-x}Bi_x$ ($x = 0.5$ and 1). The higher strain of $Mg_3Sb_{0.5}Bi_{1.498}Te_{0.002}$ can be attributed to

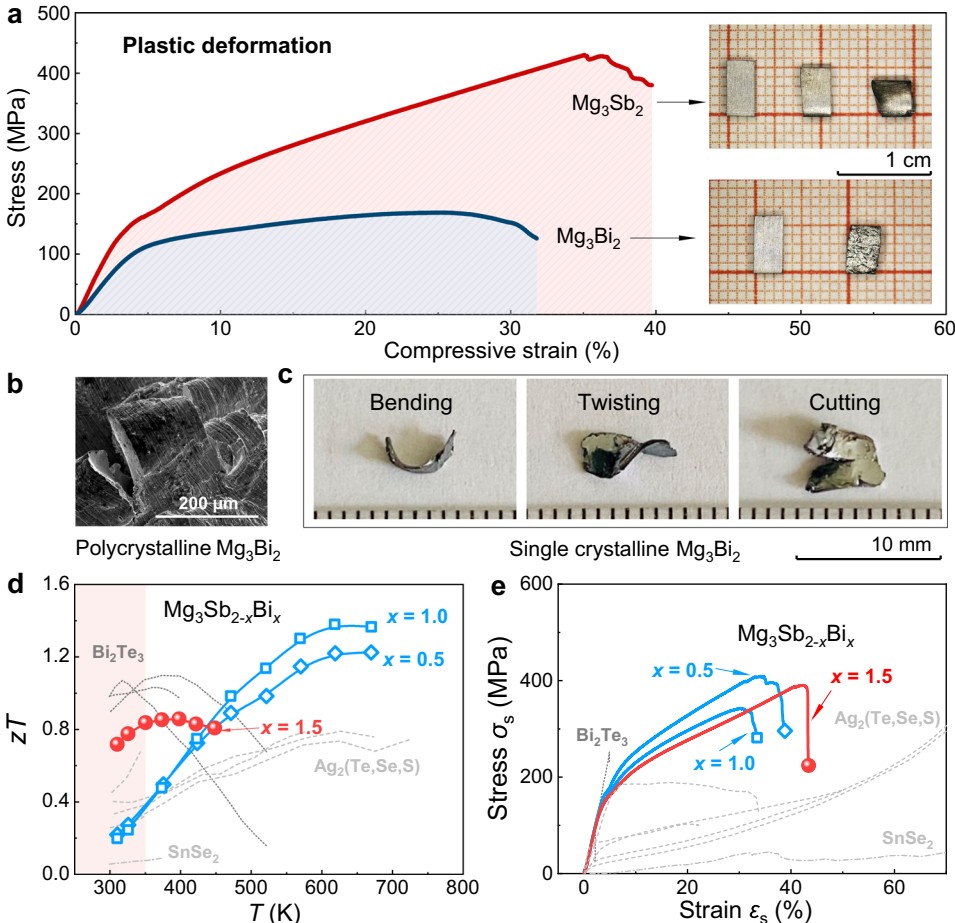

**Fig. 1 | The plasticity of Mg$_3$Sb$_{2-x}$Bi$_x$. a** Compressive stress and strain curves of polycrystalline Mg$_3$Sb$_2$ and Mg$_3$Bi$_2$ with optical images inset showing Mg$_3$Sb$_2$ and Mg$_3$Bi$_2$ after 33.3% and 16.7% compression, respectively; **b** fracture surface morphology of compressed polycrystalline Mg$_3$Bi$_2$; **c** optical images of single-crystalline Mg$_3$Bi$_2$ by bending, twisting and cutting; **d** temperature dependence of $zT$ and **e** compressive strain–stress curves of polycrystalline Te-doped Mg$_3$Sb$_{2-x}$Bi$_x$, Bi$_2$Te$_3$-based compounds, Ag$_2$(Te,Se,S) and SnSe$_2$[3,7,8,18–21,27,37,58].

the better plasticity of Mg$_3$Bi$_2$ compared to Mg$_3$Sb$_2$ as revealed above, which is consistent with the result recently reported[43]. As a result of the optimized Bi contents, a combination of high plasticity and high TE performance at room temperature has been achieved in low-cost Bi-rich Mg$_3$Sb$_{0.5}$Bi$_{1.498}$Te$_{0.002}$, which significantly surpasses both commercial Bi$_2$(Te,Se)$_3$ (strain ≤ 5%), plastic Ag$_2$(Te,Se,S) and organic TE materials ($zT$ ≤ 0.4).

It should be mentioned that Te doping has fewer effects on the plasticity of Mg$_3$Sb$_{2-x}$Bi$_x$. As shown in Supplementary Fig. 5, besides the high plasticity of Mg$_3$Sb$_{2-x}$Bi$_x$ with Te doping (Fig. 1e), p-type polycrystalline Mg$_3$Sb$_{2-x}$Bi$_x$ without doping also exhibits high plasticity (30% compressive strain) across various Sb/Bi ratios. This suggests that plasticity is an inherent characteristic of the Mg$_3$Sb$_{2-x}$Bi$_x$ system, and also implies that plasticity is less dependent on electrical transport properties. Therefore, by fine-tuning the carrier concentration and mobility, it is feasible to achieve simultaneous high TE performance and high plasticity in Mg$_3$Sb$_{2-x}$Bi$_x$, i.e., the combination of large strain (~43%) and high $zT$ (~0.72) achieved in this work via sintering, which is much better than the result reported recently via cold compression[43].

## Plastic deformation mechanism

Generally, in metals and alloys, dislocations are pivotal in facilitating atomic layer slipping, thus enabling plastic deformation. To reveal the origin of high plasticity in Mg$_3$Sb$_{2-x}$Bi$_x$ semiconductors, a microstructure study using the scanning transmission electron microscope (STEM) was conducted. Single-crystalline Mg$_3$Sb$_2$ was selected due to its high resistance against moisture in the air, favorable for efficient microstructure analysis[44]. As shown in Fig. 2a, a notable presence of dislocations can be identified, which will contribute to the high plasticity of Mg$_3$Sb$_2$. During the plastic deformation of Mg$_3$Sb$_2$, the slipping can readily occur owing to the dislocation generation and movement. In addition, an orientation difference can be observed in Mg$_3$Sb$_2$ single crystal, which should also contribute to the plasticity by providing an alternative energy-consuming mechanism, akin to twinning- and amorphization-facilitated plastic deformation in high-entropy alloys[45].

Why do the slipping and dislocations easily occur in Mg$_3$Sb$_2$ semiconductors? Recent studies on Ag$_2$S and other plastic inorganic semiconductors propose small slipping energy or generalized stacking fault energy (GSFE) and large cleavage energy (CE) as criteria for high plasticity[10]. In this study, the GSFE and CE of Mg$_3$Sb$_2$ and Mg$_3$Bi$_2$ have been calculated, as shown in Fig. 2b, c, respectively. Taking Mg$_3$Sb$_2$ as an example, both [100](001) and [110](001) directions are identified as probable slipping directions due to their small GSFE values. Considering the crystal symmetry in Mg$_3$Sb$_2$ (trigonal crystal structure with $P\bar{3}m1$ space group), there are 8 equivalent slip systems. Moreover, the maximum CE (of ~1.4 J m$^{-2}$) is twice the maximum value of GSFE along these directions (of ~ 0.7 J·m$^{-2}$), indicating ease of slipping but difficulty of cleaving. The abundance of available slip systems, coupled with small GSFE and large CE, contributes to the high plasticity of Mg$_3$Sb$_2$. Similar findings can be confirmed in Mg$_3$Bi$_2$, which also exhibits small GSFE and high CE.

Intrinsically, from a chemical bonding perspective, unlike the delocalized metallic bonds in metals, covalent and ionic bonds typically hinder plastic deformation due to localized electrons. However, in the case of inorganic semiconductors like $Mg_3Sb_2$ and $Mg_3Bi_2$, which both possess covalent bonds according to the charge density difference (CDD) in Supplementary Fig. 6, it is intriguing that they exhibit such high plasticity. Focusing on the [100](001) slipping direction of $Mg_3Sb_2$, we calculated the variation of chemical bonding during slipping. As shown in Fig. 2d, CDDs at five different relative displacements (RDs) during slipping were examined, with their corresponding projections on the (001) plane highlighted in the red dashed box. The whole CDDs with RD increasing from 0 to 1 can be found in Supplementary Figs. 7, 8.

Initially (0.0 RD), three bonds exist between Sb and Mg atoms. As RD increased slightly to 0.1, one Mg-Sb bond disappeared quickly, while the other two Mg-Sb bonds can still be observed. Further increasing RD to 0.5 results in only one remaining Mg-Sb bond, with the other two broken. Surprisingly, with a continued increase of RD, new Mg-Sb bonds are sequentially formed until returning to the initial state with three intact bonds. Consequently, despite the breakage of some Mg-Sb bonds during slipping, at least one bond remains within the slipped layers, crucial for maintaining the structural integrity of $Mg_3Sb_2$ without fracturing during deformation.

To gain a deeper understanding of these persistent bonds between Mg and Sb atoms, we calculated the integrated crystal orbital Hamilton population (ICOHP) for Mg1-Sb, Mg2-Sb, and Mg3-Sb bonds. Illustrated in Fig. 2e, as the RD increased, both Mg1-Sb and Mg2-Sb bonds swiftly break, aligning with CDD results in Fig. 2d. However, Mg3-Sb bonds, instead of breakage, undergo continuous strengthening, ensuring the retention of bonding states in the slipped layer. In conventional covalent compounds, directional chemical bonding between slipping layers typically leads to bond breakdown and crack formation during slipping, but the circumstance is quite different for $Mg_3Sb_2$. Although Mg1-Sb and Mg2-Sb bonds are broken during deformation, Mg3-Sb bonds persist and even strengthen, which maintains the structural integrity. Similar trends are observed in $Mg_3Bi_2$, where Mg1-Bi and Mg2-Bi bonds break while Mg3-Bi bonds persist and strengthen during slipping (Supplementary Figs. 9–11). Consequently, from a chemical bonding perspective, the high plasticity of $Mg_3Sb_2$ and $Mg_3Bi_2$ can be attributed to the persistent Mg3-Sb/Bi bonds in the slipped layer, which prevents structural collapse and may also facilitate the dislocation generation and movement, just as the high density of dislocations observed in STEM (Fig. 2a).

## High toughness and good machinability in $Mg_3Sb_{2-x}Bi_x$

Synthesized TE materials are typically cut or diced into TE legs, and materials with high toughness can withstand substantial energy, allowing them to be cut or diced into small dimensions with a high yield rate. Toughness can be assessed by the area under the strain-stress curve. Therefore, alongside high plasticity, high strength is also crucial for materials exhibiting high toughness. Figure 3a summarizes both TE and mechanical performances of $Mg_3Sb_{0.5}Bi_{1.498}Te_{0.002}$, $Bi_2Te_3$[8], $Ag_2(Te,S)$[20] and $SnSe_2$[21]. Among them, polycrystalline $Mg_3Sb_{0.5}Bi_{1.498}Te_{0.002}$ exhibits high $zT$, high plasticity (high strain $\varepsilon_s$), and high compressive strength $\sigma_s$ simultaneously at room temperature. This compelling combination of TE and mechanical performance makes polycrystalline $Mg_3Sb_{0.5}Bi_{1.498}Te_{0.002}$ particularly suitable for room temperature applications, especially considering its low cost. Due to the high strength and plasticity, all polycrystalline $Mg_3Sb_{2-x}Bi_x$ possesses intrinsic high toughness. As shown in Fig. 3b, polycrystalline $Mg_3Sb_{0.5}Bi_{1.498}Te_{0.002}$ displays simultaneously high TE performance and high toughness at room temperature, making it competitive among the current plastic TE materials.

Due to the high toughness resulting from both high plasticity and strength, polycrystalline $Mg_3Sb_{2-x}Bi_x$ can be diced into small sizes without damage. In contrast, commercial polycrystalline $(Bi,Sb)_2Te_3$ and $Bi_2(Te, Se)_3$ fail to be diced much smaller with numerous granules peeled off, as demonstrated in Fig. 3c (detailed dicing images with gradual cut distances from 200 μm to 10 μm can be found in Supplementary Fig. 12). For $Mg_3Sb_2$, intact granules with dimensions of about $100 \times 100\ \mu m^2$ and $50 \times 50\ \mu m^2$ can be readily obtained with a high yield rate. $Mg_3Sb_{0.5}Bi_{1.498}Te_{0.002}$ with superior TE performance can also be effectively diced with the dimension of $100 \times 100\ \mu m^2$. However, diced $(Bi,Sb)_2Te_3$ and $Bi_2(Te,Se)_3$ with dimensions of ~$150 \times 150\ \mu m^2$ show obvious damaged edges, as shown in Supplementary Fig. 13. Recently, micro-TE devices made of micrometer TE legs have gained much attention, especially for potential applications in 5G communications. The high yield rate of smaller dimension TE legs in $Mg_3Sb_{2-x}Bi_x$ will also benefit its development in micro-TE modules.

## Prototype flexible TE modules based on $Mg_3Sb_{2-x}Bi_x$

Due to the high toughness demonstrated above, bulk $Mg_3Sb_{2-x}Bi_x$ can be processed into TE legs with varied and small dimensions (Supplementary Fig. 14) and exhibit possible flexibility if the thickness is thin enough[46]. Considering its high TE performance and high plasticity, thin $Mg_3Sb_{2-x}Bi_x$ TE legs are very suitable to be fabricated as flexible TE modules. Here, we fabricate and demonstrate prototype flexible in-plane (Fig. 4a) and out-of-plane (Fig. 4b) TE modules based on high-performance $Mg_3Sb_{0.5}Bi_{1.498}Te_{0.002}$. These prototype TE modules are all assembled using flexible polyimide (PI) film bases, Cu conducting wires and $Mg_3Sb_{0.5}Bi_{1.498}Te_{0.002}$ TE legs.

The in-plane TE module consists of nine n-type $Mg_3Sb_{0.5}Bi_{1.498}Te_{0.002}$ legs, exhibiting a maximum output voltage $V$ of ~6.2 mV and a maximum power density $P/A$ of ~0.24 μW·cm$^{-2}$ when the measured temperature difference of the module $\Delta T_{module}$ is ~5.6 K (Fig. 4c), where the $P$ is the output power, $A$ is the area of the TE module. Considering the length of the TE legs $L$, the obtained normalized power density $P \times L/A$ of this in-plane TE module reaches 14.4 μW·m$^{-1}$, significantly surpassing that of PEDOT and Poly[$A_x$(Mett)]-based organic flexible TE modules[27]. However, compared to the state-of-art in-plane flexible TE modules based on $Ag_2S_{0.5}Se_{0.5}$[27], the performance of this $Mg_3Sb_{0.5}Bi_{1.498}Te_{0.002}$ flexible TE module is relatively inferior, primarily due to the substantial internal resistance, about 27 Ω in $Mg_3Sb_{2-x}Bi_x$ against 9 Ω in $Ag_2S_{0.5}Se_{0.5}$[27]. The high internal resistance in $Mg_3Sb_{0.5}Bi_{1.498}Te_{0.002}$ flexible TE module arises from the significant contact resistance between $Mg_3Sb_{0.5}Bi_{1.498}Te_{0.002}$ and electrodes. As shown in Supplementary Fig. 15, the contact resistance between $Mg_3Sb_{0.5}Bi_{1.498}Te_{0.002}$ and Cu electrode is very high, of about 3500 μΩ·cm$^2$. Given the excellent room-temperature $zT$ of plastic $Mg_3Sb_{0.5}Bi_{1.498}Te_{0.002}$, the reduction of the interfacial resistance of the flexible TE module holds the promise of substantially enhanced performance and fosters future applications in flexible electronics.

In the case of the out-of-plane TE module, it consists of 8 $Mg_3Sb_{0.5}Bi_{1.498}Te_{0.002}$/Cu pairs. A maximum $V$ of 0.3 mV and $P/A$ of 3.9 nW·cm$^{-2}$ are achieved with $\Delta T_{module}$ of 13.1 K (Fig. 4d). However, it is notable that the performance of the out-of-plane TE module is significantly inferior to that of the in-plane module. This is primarily attributed to the small effective temperature difference $\Delta T_{leg}$ established along the thickness direction of TE legs. The $\Delta T_{leg}$ can be calculated by using the output voltage of the modules and the Seebeck coefficient of the TE material[3]. Specifically, the calculated $\Delta T_{leg}$ is 3.1 K for the in-plane TE module (with a measured $\Delta T_{module}$ of 5.6 K), whereas it is only 0.17 K for the out-of-plane TE module (with a measured $\Delta T_{module}$ of 13.1 K). Considering this $\Delta T_{leg}$ for the calculation of the normalized power density $P/(A \times \Delta T^2)$ proposed by ref. 3, it reaches 0.13 μW·cm$^{-2}$·K$^{-2}$, which surpasses nearly all organic-based flexible TE modules but falls short of $Ag_2S$-based flexible TE modules[3], primarily due to the much higher internal resistance ~ 7 Ω. It is worth mentioning that regardless of whether it is an in-plane or out-of-plane

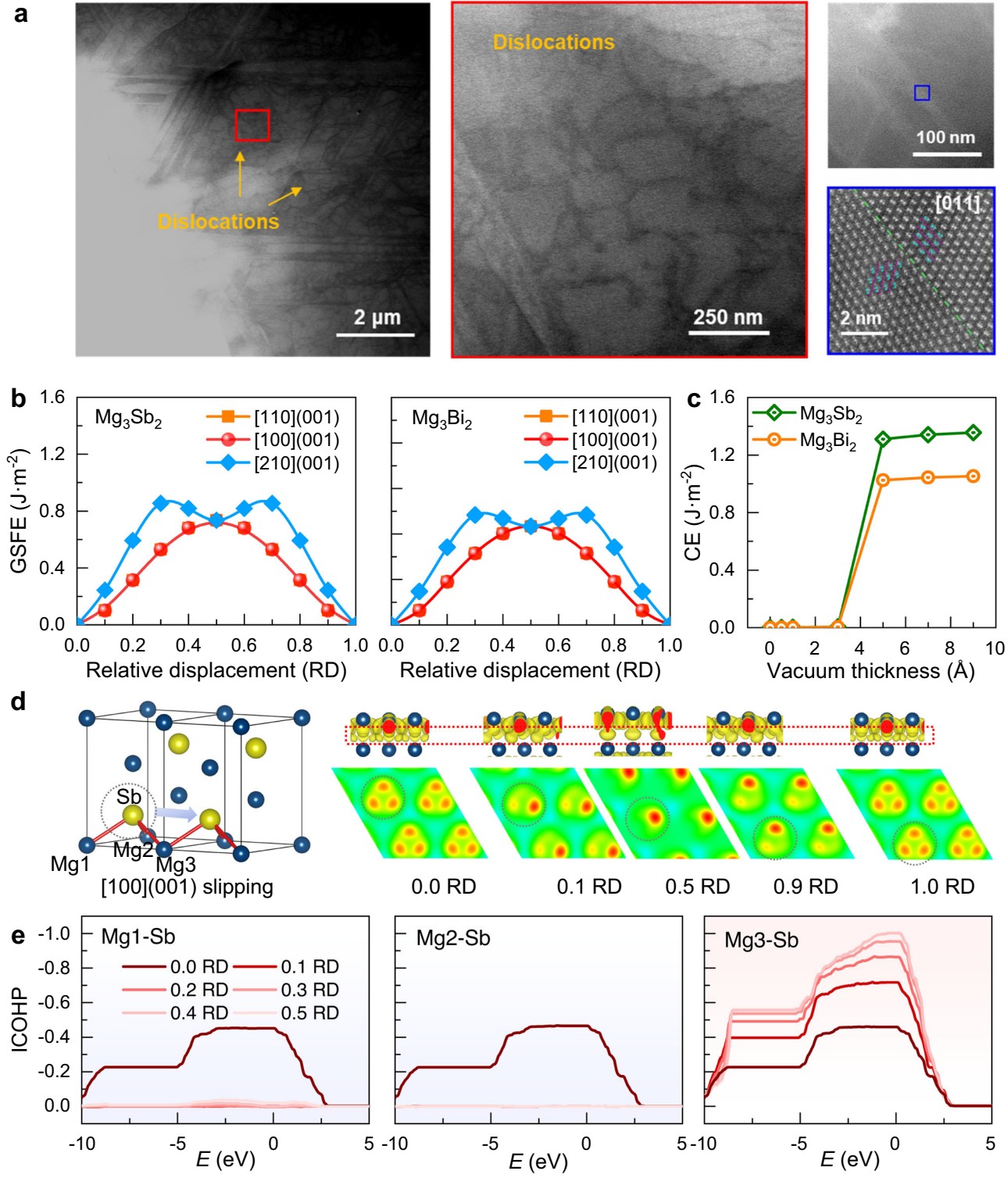

**Fig. 2 | The plastic deformation mechanism of Mg₃Sb₂₋ₓBiₓ. a** STEM images of single-crystalline $Mg_3Sb_2$ revealing dislocations and different crystal orientations; **b** the GSFE of $Mg_3Sb_2$ and $Mg_3Bi_2$; **c** the CE of $Mg_3Sb_2$ and $Mg_3Bi_2$; **d** the CDDs between Mg and Sb atoms in the slipped layer; **e** the ICOHPs for Mg1-Sb, Mg2-Sb and Mg3-Sb bonds with different relative displacements RDs.

TE module made of $Mg_3Sb_{0.5}Bi_{1.498}Te_{0.002}$, the interface between materials and electrodes remains the principal obstacle limiting the module's high performance, despite the materials exhibiting high TE performance. Additionally, in previous Ag-based flexible TE modules, high-performance p-type AgCuSe-based materials were used[3]. This also suggests possible ways to future improve the output performance of $Mg_3Sb_{2-x}Bi_x$ flexible modules if high-performance p-type Mg-based TE materials are developed. Optimization of the interface, such as using appropriate interface materials[47,48], and advancement of high-performance p-type plastic TE materials hold promise for achieving significantly better performance of the flexible TE module based on $Mg_3Sb_{2-x}Bi_x$.

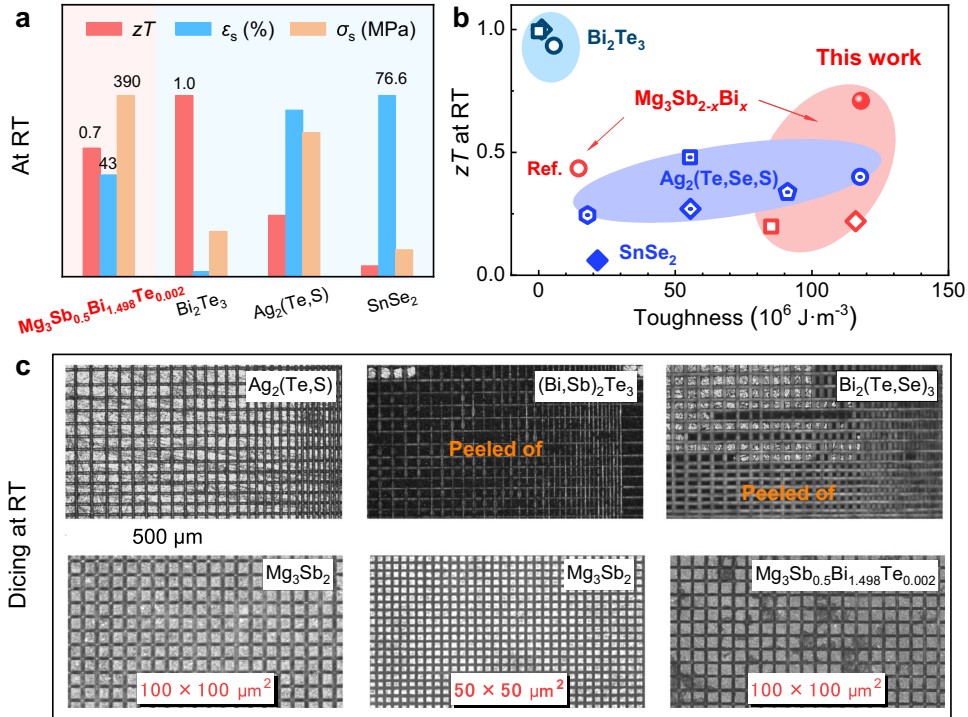

**Fig. 3 | The mechanical performance, TE performance and machinability of polycrystalline $Mg_3Sb_{2-x}Bi_x$. a** The summarized $zT$, $\varepsilon_s$ and $\sigma_s$ of $Mg_3Sb_{0.5}Bi_{1.498}Te_{0.002}$, $Bi_2Te_3$-based compounds[8], $Ag_2(Te,S)$[20] and $SnSe_2$[21] at room temperature; **b** the room temperature $zT$ vs. toughness in $Mg_3Sb_{2-x}Bi_x$, $Bi_2Te_3$-based compounds[3,7,8,18–21,27,37,58], $Ag_2(Te,Se,S)$ and $SnSe_2$[3,7,8,18–21,27,37,58]; **c** the optical images of diced $Ag_2(Te,S)$, $(Bi,Sb)_2Te_3$, $Bi_2(Te,Se)_3$, $Mg_3Sb_2$ and $Mg_3Sb_{0.5}Bi_{1.498}Te_{0.002}$.

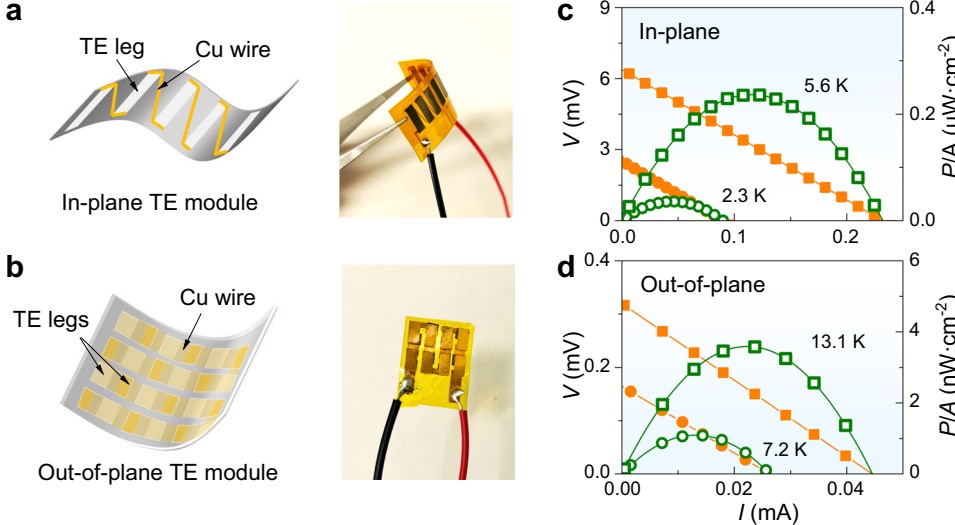

**Fig. 4 | The in-plane and out-of-plane TE modules of $Mg_3Sb_{2-x}Bi_x$.** The fabricated **(a)** in-plane and **(b)** out-of-plane TE module based on $Mg_3Sb_{0.5}Bi_{1.498}Te_{0.002}$ with its schematics on the left; the output voltage $V$ and power density $P/A$ of **(c)** in-plane and **(d)** out-of-plane TE modules. The temperature value is the temperature difference of the module $\Delta T_{module}$.

## Discussion

In this study, the inherent high plasticity of $Mg_3Sb_2$ and $Mg_3Bi_2$ has been revealed. Despite intrinsic covalent/ionic bonding, polycrystalline $Mg_3Sb_2$ and $Mg_3Bi_2$ hold over 30% compressive strain. By optimizing the Bi contents, compressive strain ~43% and $zT$ ~ 0.72 can be achieved simultaneously in Bi-rich polycrystalline $Mg_3Sb_{0.5}Bi_{1.498}Te_{0.002}$ at room temperature, surpassing the room temperature performance of known plastic TE semiconductors. The persistent Mg-Sb/Bi bonds within the slipped layer, along with the abundance of dislocations, are revealed as important contributors to

the high plasticity of $Mg_3Sb_{2-x}Bi_x$, ensuring structural integrity maintenance and facilitating atomic layer slipping during deformation, respectively.

Furthermore, the high toughness resulting from the high plasticity and strength enables polycrystalline $Mg_3Sb_{2-x}Bi_x$ to be easily diced into granules below $100 \times 100\ \mu m^2$ without edge damage and cut into TE legs of various dimensions. Both in-plane and out-of-plane flexible TE modules based on polycrystalline $Mg_3Sb_{0.5}Bi_{1.498}Te_{0.002}$ have been assembled and demonstrated, exhibiting much higher performance compared to organic-based flexible TE modules with promising

opportunities for further enhancement by reducing the interface resistance. The revealed intrinsically high plasticity, high TE performance as well as good machinability in polycrystalline $Mg_3Sb_{2-x}Bi_x$ will advance their potential applications in flexible electronics.

## Methods

### Materials synthesis

$Mg_3Sb_{2-x}Bi_x$ and $Mg_3Sb_{2-x}Bi_{x-y}Te_y$ ($x = 0.5, 1.0, 1.5$; $y = 0.002, 0.01, 0.02$) were synthesized by mechanical alloying with 10% excessive Mg contents. Mg powders (99.8%), Sb powers (99.9%), Bi powders (99.9%) and/or Te powders (99.999%) were used and weighed in the glove box, and then mechanically alloyed for 2 h (SPEX-8000D, PYNN). The obtained powder products were solidated by vacuum spark plasma sintering (LABOX-650F, Sinter Land Inc.) under 973 K and 60 MPa for 2 min. The relative density of all sintered samples reaches above 97%. The single-crystalline $Mg_3Sb_2$ and $Mg_3Bi_2$ were grown by self-flux method with excess Sb/Bi[44,49]. The $Ag_2(Te,S)$ used for dicing tests were obtained by melting method. High-purity Ag shots (99.999%), Te shots (99.999%), and S flakes (99.999%) were used and weighted accordingly, which were then loaded and sealed in the quartz tube for the melting at 1273 K[19], and the commercial $Bi_2(Te,Se)_3$ and $(Bi,Sb)_2Te_3$ were produced by hot extrusion.

### Characterization and measurements

The compressed fracture surface morphology was investigated by the SEM (Hitachi, S-3400N). The microstructures of $Mg_3Sb_2$ single crystal were studied by high-resolution STEM (Cs corrected JEOL ARM 200F microscope). The TE performance $zT$ of $Mg_3Sb_{2-x}Bi_{x-y}Te_y$ was calculated by the formula: $zT = S^2\sigma T/\kappa$, in which the $S$ and the $\sigma$ were measured by Linseis LSR-3 system with measurement uncertainties of $S$ and $\sigma$ about ±5% and ±3%, respectively, and the $\kappa$ was calculated by the formula: $\kappa = D\rho C_p$, where the thermal diffusivity $D$ was measured by Netzsch LFA457 with about ±3% uncertainty, the sample density $\rho$ was estimated by the Archimedes method, and heat capacity $C_p$ was calculated according to a previous study[50]. The calculated uncertainty of $zT$ is within ±10%. Compressive tests of cuboids ($3 \times 3 \times 6$ mm$^3$) and tensile tests of plates ($3 \times 0.7 \times 27$ mm$^3$) were performed by a universal testing machine (Siomm, JVJ-20S, China) under a loading rate of 1 mm·min$^{-1}$. The experiments of dicing were carried out by using a commercialized dicing machine (Qisheng-D0620, China) with a spindle speed of about 30,000 rpm and a dicing speed of 0.3 mm/s. The dicing did not cut the ingot thoroughly, which allowed the cut blocks to remain attached to the original ingot. Two types of cutting programs have been used. The first is a regular cut, where the ingot is cut into blocks of $150 \times 150$ μm$^2$, $100 \times 100$ μm$^2$ or $50 \times 50$ μm$^2$. The second is a gradual cut, where the cut distance decreases in every two cut steps: 200 μm, 150 μm, 100 μm, 50 μm, 20 μm, and finally 10 μm.

### Module fabrication and test

High-performance $Mg_3Sb_{0.5}Bi_{1.498}Te_{0.002}$ TE legs with dimensions of $1.8 \times 0.12 \times 6$ mm$^3$ and $1.8 \times 1.8 \times 0.12$ mm$^3$ were used for fabricating in-plane and out-of-plane TE modules, respectively. The thickness of $Mg_3Sb_{0.5}Bi_{1.498}Te_{0.002}$ TE legs is first reduced by cutting to 0.5 mm and then by manually polishing to 0.12 mm. PI films and Cu sheets were used as support bases and conducting wires. High conductive Ag pastes were used to connect TE legs and Cu wires. Contact resistance between $Mg_3Sb_{0.5}Bi_{1.498}Te_{0.002}$ and Cu joint was measured by using a home-build instrument[51]. A home-build instrument with a source meter (K2400, Keithley) was used to measure the output voltage and source current of the TE modules. The temperature gradient is applied along the length direction in the in-plane module and thickness direction in the out-of-plane module by one-side heating and the other-side cooling. Two K-type thermocouples were used to record the temperatures of the hot and cold sides of the module, respectively. $\Delta T_{module}$ was obtained by the differences of the temperatures measured by the two thermocouples, while $\Delta T_{leg}$ was calculated by using the output voltage of the modules and the Seebeck coefficient of the TE material[3].

### First-principles calculations

First-principles calculations were performed by the software Vienna ab initio Simulation Package (VASP) with the projector augmented-wave method based on density functional theory[52,53]. Generalized gradient approximation - Perdew-Burke-Ernzerhof type (GGA-PBE) and modified Becke-Johnson were used as the exchange-correlation functionals[54,55]. Plane-wave energy cutoff, Hellmann-Feynman force on each atom energy and convergence criterion were set as 500 eV, 0.001 eV·Å$^{-1}$ and $10^{-8}$ eV, respectively. Geometry relaxation and self-consistent static calculations adopted the Gamma-centered $k$-point sampling with $k = 30/L$ and $60/L$, respectively, where the $L$ is the corresponding lattice parameter. The GSFE and CE were calculated based on the $2 \times 2 \times 4$ supercell. To analyze the GSFE, $2 \times 2 \times 2$ half of the supercell is artificially shifted along a specific crystallographic direction at 10-step RDs. For CE analysis, a vacuum layer with varied thickness is inserted into the half of the supercell to simulate the artificial separation of the crystal into two parts. VASPKIT[56] and Lobster[57] have been used to post-process the calculated data, including CDD and ICOHP.

## Data availability

All data generated or analyzed during this study are included in this published article (and its supplementary information file).

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

## Acknowledgements

This work was supported by the National Key Research and Development Program of China (No. 2023YFB3809400, T.Z.), the National Natural Science Foundation of China (No. U23A20553, T.Z.; No. 92163203, T.Z.; No. 52101275, C.F.), the Fundamental Research Funds for the Zhejiang Provincial Universities (No. 226-2023-00001, C.F.).

## Author contributions

A.L., C.F. and T.Z. designed the project. A.L. and Y.W. prepared the samples and carried out the transport measurements, mechanical tests and first-principles calculations. Y.W., Y.L. and X.Y. provided the samples of $Ag_2(Te,S)$, $Bi_2(Te,Se)_3$ and $(Bi,Sb)_2Te_3$. P.N. and B.G. carried out the scanning transmission electron microscopy study. A.L. fabricated the flexible thermoelectric modules and measured the performance with input from K.L. A.L. analyzed the data and wrote the original manuscript with input from Y.W. and C.F. C.F. and T.Z. supervised the whole project. All the authors reviewed and edited the manuscript.

## Competing interests

The authors declare no competing interests.
