## [Peer Review File · Nature Communications]

High performance magnesium-based plastic semiconductors for flexible thermoelectricsREVIEWER COMMENTS

Reviewer #1 (Remarks to the Author):

The authors investigate the toughness and plasticity of Mg₃Sb₂, Mg₃Bi₂, and polycrystalline Mg₃Sb_{2-x}Bi_x at room temperature. Thermoelectric properties of Mg₃Sb_{0.5}Bi_{1.498}Te_{0.002} are reported. A thermoelectric module made of Mg₃Sb_{2-x}Bi_x is made and tested. The conclusions are supported by the results and analysis. The article is interesting and relevant for the topic. It is well written, but at times it can be challenging to follow what exactly the authors did and why. Some clarifications and further explanations to help the reader follow the thought process and better understand the conclusions would be appreciated. Following suggestions should also be considered:

- 1) The fracture morphology of single crystal Mg₃Sb₂ and Mg₃Bi₂ after compression is mentioned (row 121). Have similar fractures been observed when measuring strain (row 137 and 138) for polycrystalline Mg₃Sb_{2-x}Bi_x? Is plasticity linearly dependent on the maximum strain before fracture? Are there any other factors to consider?
- 2) When comparing to other materials, be clear which materials you are comparing to. For example: "As shown in Fig. 3b, polycrystalline Mg₃Sb_{0.5}Bi_{1.5} displays double high TE performance and toughness at room temperature, making it extremely competitive among the current plastic TE materials" (row 246) – if both TE and toughness are double compared to other plastic materials, then it should be superior, not only competitive. Are there any other considerations? That is a strong claim to make – is that the case for all other plastic TE materials?
- 3) In figure 2c, what is meant by vacuum thickness?
- 4) In figure 3, use "at" instead of "@". In figure 3b, why are there two kinds of symbols for this work (red filled circle and a star)?
- 5) The out of plane temperature difference is estimated to be 0.17 K. How is it estimated so precisely? It would be helpful if the estimated uncertainties of all measurements were clearly stated or displayed in figures in the form of error bars, if possible.
- 6) "using thin Mg₃Sb_{2-x}Bi_x TE legs with a thickness of 120 μm" (row 277) – why has this thickness been chosen? In the same paragraph, it is mentioned that flexibility is exhibited, if the thickness is "thin enough" – is 120 μm thin enough?
- 7) It is stated that the performance of Mg₃Sb_{2-x}Bi_x flexible module is inferior to the Ag-based one (row 286). In the abstract it is mentioned that the Mg₃Sb_{2-x}Bi_x material surpasses the TE performance of Ag-based materials at room temperature (row 25). Is internal resistance the only reason?

Reviewer #2 (Remarks to the Author):

In this work, the authors have achieved high thermoelectric performance (figure of merit ZT ~ 0.72) and high ductility (strain ~ 43%) in Mg₃Sb_{2-x}Bi_x by optimising the proportions of magnesium, bismuth, and antimony at room temperature. This is a notable highlight of the research. It is particularly commendable that the authors have provided a detailed discussion of the mechanisms behind ductility formation, which will assist in further understanding and development of new flexible thermoelectric materials. The paper also preliminarily demonstrates the prototype manufacturing of flexible thermoelectric modules based on these materials, showcasing their potential application in real devices, especially in wearable and flexible electronics. However, there are some shortcomings in the paper, such as in the

explanation of details, exploration of scientific bases, and completeness of testing. It is this reviewer's opinion that the manuscript could be accepted after major revisions. The authors are advised to consider the following review comments to further improve the quality of the manuscript.

1. The initial section of the manuscript presents a discussion of the continuous compression of polycrystalline Mg₃Bi₂ up to 1.6 mm with a strain of 80%. Although this data is primarily intended to demonstrate the mechanical properties and ductility of polycrystalline Mg₃Bi₂, it might lead readers to mistakenly compare it with previous data on polycrystalline Mg₃Sb₂, potentially leading to misestimations of the compressive performance of Mg₃Bi₂. It is therefore recommended that data equivalent in meaning to that of the compression tests on polycrystalline Mg₃Sb₂ be provided here for a better comparison. Furthermore, it is recommended that the compression tests on polycrystalline Mg₃Bi₂ and the tensile tests on polycrystalline Mg₃Sb₂ be conducted in parallel, with both the tensile tests on Mg₃Bi₂ and the compression tests on Mg₃Sb₂ being comprehensively reported in sections S1 and S2.
2. In Figure 1e, the two blue lines lack specific labelling.
3. In the discussion of the high thermoelectric performance of Mg₃Sb₂-xBi_x, although the Seebeck coefficient, electrical conductivity, and thermal conductivity were tested, there was no further analysis from a perspective on the reasons for changes in material performance parameters. It is recommended that analysis of the electronic structure and carrier mobility be included, which would help in understanding the reasons behind the improvements in thermoelectric performance more comprehensively.
4. Figure S13 demonstrates that the cut small blocks exhibit areas of damage. However, it is unclear whether the author placed these cut blocks within a grid for analysis, as the information in this figure is unclear and not intuitive. Additionally, it is recommended that the quality of images in Figures 3 and S12 be improved. Although Figure S12 includes a scale, it would be beneficial to additionally note the dimensions of the cut blocks for comparison. Furthermore, it would be helpful to clarify whether the small blocks shown at the bottom of Figure 3c are of the same dimensions as those in the images above.
5. In the discussion of the high internal resistance of the flexible TE module of Mg₃Sb₂-xBi_x, although high internal resistance is attributed to significant interfacial resistance between Mg₃Sb₂-xBi_x and the electrodes, the actual test results were not provided. It is therefore necessary to include interfacial resistance tests between Mg₃Sb₂-xBi_x and the electrodes.
6. Similarly, when discussing the suboptimal out-of-plane performance of the thermoelectric module, which is primarily attributed to the magnitude of the temperature differential, this component should also utilize thermocouples for precise temperature measurement and analysis in order to identify the variables influencing temperature.

Reviewer #3 (Remarks to the Author):

Flexible thermoelectrics could be a potential sustainable power supply for flexible electronics. In this work, Li et al. report a high-performance Mg-based plastic semiconductor. By revealing the intrinsic plasticity in Mg₃Sb₂ and Mg₃Bi₂, and through adjusting the Sb/Bi ratios in Mg₃Sb₂-xBi_x semiconductors, they attained a high zT value of ~0.72 at room temperature, as well as a large compressive strain of 43%, in polycrystalline Mg₃Sb_{0.5}Bi_{1.498}Te_{0.002}. Based on this high-performance plastic thermoelectric semiconductor, they fabricated both prototype in-plane and out-of-plane flexible thermoelectric modules, demonstrating the potential of this material for flexible electronics. In my opinion, this work is systematic, interesting, and cutting-edge in the field. Previously, flexible thermoelectrics were generally developed based on organic semiconductors or

inorganic/organic hybrids with inferior thermoelectric performance. The finding of room-temperature plasticity in low-cost $\text{Mg}_3\text{Sb}_{2-x}\text{Bi}_x$ inorganic semiconductors with good thermoelectric performance and the fabricated flexible modules will advance the future development of flexible thermoelectric technology using high-performance inorganic semiconductors. I would like to recommend this work for publication in Nature Communications after some minor revisions.

1. The single-crystalline Mg_3Bi_2 displays good deformability when subjected to bending and twisting in Fig. 1c. How about the single-crystalline Mg_3Sb_2 ?
2. The flexible thermoelectric modules are assembled by using thin legs that were cut from the ingots. Compared to the “bottom-up” fabrication technology, i.e. ink direct-writing and sputtering, what are the advantages of the fabrication method in this work?
3. The authors stated the inferior output performance of the flexible module is a result of the high internal resistance. What are the reasons for this high resistance? Are there some ways to further reduce the internal resistance?
4. Besides the flexible power generator, could the plastic $\text{Mg}_3\text{Sb}_{2-x}\text{Bi}_x$ semiconductors be used for other functions in flexible electronics?
5. The scale bar in Fig. 1c is missing.
6. Fig. 3c, since different processes could result in varied mechanical properties, the methods for synthesizing $\text{Ag}_2(\text{Te},\text{S})$, $(\text{Bi},\text{Sb})_2\text{Te}_3$, and $\text{Bi}_2(\text{Te},\text{Se})_3$ should be mentioned.

Dear Reviewers,

Thank you very much for your valuable comments and suggestions on our manuscript, which really helps improve the quality of this work. The point-to-point response is listed below and the manuscript is revised accordingly. We hope that now the revised manuscript could be suitable for publication.

Sincerely yours,

Tiejun Zhu

Reviewer #1 (Remarks to the Author):

Comment: 1) The fracture morphology of single crystal Mg_3Sb_2 and Mg_3Bi_2 after compression is mentioned (row 121). Have similar fractures been observed when measuring strain (row 137 and 138) for polycrystalline $\text{Mg}_3\text{Sb}_{2-x}\text{Bi}_x$? Is plasticity linearly dependent on the maximum strain before fracture? Are there any other factors to consider?

Response: Thank you very much for your questions. As seen in Fig. 1a and Fig. 1e, all the polycrystalline $\text{Mg}_3\text{Sb}_{2-x}\text{Bi}_x$ samples demonstrate similar deformation behavior, i.e., first experiencing elastic deformation, then plastic deformation and ultimately fracture. We have examined the fracture morphologies of all polycrystalline $\text{Mg}_3\text{Sb}_{2-x}\text{Bi}_x$ samples. As shown in Fig. R1, all of them display sharp cracks upon fracturing. Meanwhile, it can be noticed that the polycrystalline $\text{Mg}_3\text{Sb}_{0.5}\text{Bi}_{1.5}$ has similar fracture morphologies to Mg_3Bi_2 . We have updated the related text for this information: "...All polycrystalline $\text{Mg}_3\text{Sb}_{2-x}\text{Bi}_x$ samples display cracks upon finally fracturing. It can also be noticed that the fracture morphology of polycrystalline Bi-rich $\text{Mg}_3\text{Sb}_{2-x}\text{Bi}_x$ ($x = 1.5$) closely resembles that of Mg_3Bi_2 (Supplementary Fig. S3)...".

The maximum strain can reflect the plasticity of a material. However, the relationship between plasticity and maximum strength before fracture is less straightforward. This is because maximum strength is influenced by various factors, including, but not limited to, alloying effects (such as Bi/Sb in this study), defects and the inherent strength of the matrix phase (notably, the different strengths of Mg_3Sb_2 and Mg_3Bi_2 in this study).

Fig. R1 Fracture surface morphology of polycrystalline $\text{Mg}_3\text{Sb}_{2-x}\text{Bi}_x$ after compression.

Comment: 2) When comparing to other materials, be clear which materials you are comparing to. For example: “As shown in Fig. 3b, polycrystalline $\text{Mg}_3\text{Sb}_{0.5}\text{Bi}_{1.5}$ displays double high TE performance and toughness at room temperature, making it extremely competitive among the current plastic TE materials” (row 246) – if both TE and toughness are double compared to other plastic materials, then it should be superior, not only competitive. Are there any other considerations? That is a strong claim to make – is that the case for all other plastic TE materials?

Response: Thank you for your comments and suggestions. We apologize for the misunderstanding of the word “double”. In this text, “double high” does not mean that

Mg₃Sb_{0.5}Bi_{1.5} has twice the toughness and zT value. Instead, it refers to having two distinct high-performance characteristics: higher toughness comparable to Bi₂Te₃-based compounds and higher zT values comparable to Ag₂S and SnSe₂-based compounds. To avoid misunderstanding, we have deleted the word “double” in the relevant sentence.

Comment: 3) In figure 2c, what is meant by vacuum thickness?

Response: Thank you for your question. We used vacuum thickness to set up the crystal model for analyzing cleavage energy. When a crystal is cleaved, it splits into two parts. To simulate this, the crystal was separated by inserting a vacuum layer of varying thickness, referred to as the “vacuum thickness”. Similar notifications can also be found in Ref. 16. To make it more understandable, we have added more details in the Methods section: “...To analyze the GSFE, $2 \times 2 \times 2$ half of the supercell is artificially shifted along a specific crystallographic direction at 10-step RDs. For CE analysis, a vacuum layer with varied thickness is inserted into the half of the supercell to simulate the artificial separation of the crystal into two parts....”

Comment: 4) In figure 3, use “at” instead of “@”. In figure 3b, why are there two kinds of symbols for this work (red filled circle and a star)?

Response: Thank you for your suggestions. We have replaced “@” with “at”. We apologize for the confusion regarding the symbols in Fig. 3b, we have updated the symbols to ensure their consistency with Figs. 1d and 1e.

Comment: 5) The out of plane temperature difference is estimated to be 0.17 K. How is it estimated so precisely? It would be helpful if the estimated uncertainties of all measurements were clearly stated or displayed in figures in the form of error bars, if possible.

Response: Thank you very much for your careful reading. We calculated the temperature difference ΔT_{leg} along the TE legs by using the output voltage of the module and the Seebeck coefficient of the material, as used in the literature. To clarify,

we have added more details on ΔT_{leg} in the relevant section: "...The ΔT_{leg} can be calculated by using the output voltage of the modules and the Seebeck coefficient of the TE material³...". We have also added the measurement uncertainties in the Methods sections.

Comment: 6) "using thin $\text{Mg}_3\text{Sb}_{2-x}\text{Bi}_x$ TE legs with a thickness of $120\ \mu\text{m}$ " (row 277) – why has this thickness been chosen? In the same paragraph, it is mentioned that flexibility is exhibited, if the thickness is "thin enough" – is $120\ \mu\text{m}$ thin enough?

Response: This is a good question. It resulted from our fabrication process which involved initially cutting to $0.5\ \text{mm}$ and then manually polishing to $0.12\ \text{mm}$. It is challenging to further reduce the thickness manually, and we reduced the thickness of the TE legs as much as possible. To clarify, we have revised some expressions and added more fabrication details in Methods section: "...The thickness of $\text{Mg}_3\text{Sb}_{0.5}\text{Bi}_{1.498}\text{Te}_{0.002}$ TE legs is first reduced by cutting to $0.5\ \text{mm}$ and then by manually polishing to $0.12\ \text{mm}$...".

Comment: 7) It is stated that the performance of $\text{Mg}_3\text{Sb}_{2-x}\text{Bi}_x$ flexible module is inferior to the Ag-based one (row 286). In the abstract it is mentioned that the $\text{Mg}_3\text{Sb}_{2-x}\text{Bi}_x$ material surpasses the TE performance of Ag-based materials at room temperature (row 25). Is internal resistance the only reason?

Response: Thank you for your insightful questions. We believe the lower output performance of the $\text{Mg}_3\text{Sb}_{2-x}\text{Bi}_x$ flexible modules is primarily due to the high internal resistance. This internal resistance is a result of significantly high contact resistance between $\text{Mg}_3\text{Sb}_{0.5}\text{Bi}_{1.498}\text{Te}_{0.002}$ TE legs and copper electrodes. Additionally, we have also noticed the differences in p-type legs used in our study compared to the Ag-based flexible modules. In our work, Cu wires were used for connecting the TE legs, while high-performance AgCuSe-based p-type legs are used to match the n-type Ag-based ones. This means exploring high-performance p-type $\text{Mg}_3\text{Sb}_{2-x}\text{Bi}_x$ could contribute to higher output performance of the $\text{Mg}_3\text{Sb}_{2-x}\text{Bi}_x$ flexible modules. To address this, we have added more discussions: "...Additionally, in previous Ag-based flexible TE

modules, high-performance p-type AgCuSe-based materials were used³. This also suggests possible ways to future improve the output performance of Mg₃Sb_{2-x}Bi_x flexible modules if high-performance p-type Mg-based TE materials are developed...”.

Reviewer #2 (Remarks to the Author):

Comment: In this work, the authors have achieved high thermoelectric performance (figure of merit $ZT \sim 0.72$) and high ductility (strain $\sim 43\%$) in Mg₃Sb_{2-x}Bi_x by optimising the proportions of magnesium, bismuth, and antimony at room temperature. This is a notable highlight of the research. It is particularly commendable that the authors have provided a detailed discussion of the mechanisms behind ductility formation, which will assist in further understanding and development of new flexible thermoelectric materials. The paper also preliminarily demonstrates the prototype manufacturing of flexible thermoelectric modules based on these materials, showcasing their potential application in real devices, especially in wearable and flexible electronics. However, there are some shortcomings in the paper, such as in the explanation of details, exploration of scientific bases, and completeness of testing. It is this reviewer’s opinion that the manuscript could be accepted after major revisions. The authors are advised to consider the following review comments to further improve the quality of the manuscript.

Response: Thank you for your thorough and positive review of our work. We appreciate your questions, comments and constructive suggestions.

Comment: 1. The initial section of the manuscript presents a discussion of the continuous compression of polycrystalline Mg₃Bi₂ up to 1.6 mm with a strain of 80%. Although this data is primarily intended to demonstrate the mechanical properties and ductility of polycrystalline Mg₃Bi₂, it might lead readers to mistakenly compare it with previous data on polycrystalline Mg₃Sb₂, potentially leading to misestimations of the compressive performance of Mg₃Bi₂. It is therefore recommended that data equivalent in meaning to that of the compression tests on polycrystalline Mg₃Sb₂ be provided here

for a better comparison. Furthermore, it is recommended that the compression tests on polycrystalline Mg_3Bi_2 and the tensile tests on polycrystalline Mg_3Sb_2 be conducted in parallel, with both the tensile tests on Mg_3Bi_2 and the compression tests on Mg_3Sb_2 being comprehensively reported in sections S1 and S2.

Response: Thank you for your comments and suggestions. We apologize for the confusion that the previous data presentation incurs. In fact, the compressive tests for both polycrystalline Mg_3Sb_2 and Mg_3Bi_2 were conducted and are shown in Fig. 1a, where compressive strain over 30% in both materials can be observed. In this figure, we only present the data of Mg_3Bi_2 up to a compressive strain of 30%, because a noticeable drop in stress occurs just when the compressive strain is around 30%. Unlike Mg_3Sb_2 , which breaks suddenly with a sharp drop of the stress, Mg_3Bi_2 can still be compressed after the first noticeable drop of the stress (near the compressive strain of 30%). We presented the full compressive data for Mg_3Bi_2 in Fig. S1, which shows that Mg_3Bi_2 can be compressed to about 1.6 mm (suggesting a compressive strain of about 80%) after experiencing two noticeable drops in the stress. However, upon examining the optical image of Mg_3Bi_2 after compression, it can be seen that Mg_3Bi_2 bulk shatters into some small pieces. Thus, it is not convincing to take this high compressive strain of 80% as the true compressive performance of Mg_3Bi_2 . Given that, we only included the strain data up to 30% in Fig. 1a, while the full compressive data for Mg_3Bi_2 was shown in Fig. S1. To avoid confusion, we have revised relevant figures and sentences: "...It should be mentioned that unlike Mg_3Sb_2 , which breaks suddenly with a noticeable drop of the stress, while Mg_3Bi_2 can still be compressed after the first noticeable drop of the stress... Thus, it is not convincing to take this high compressive strain of 80% as the true compressive performance of Mg_3Bi_2 ...".

Additionally, as suggested we conducted tensile tests on both Mg_3Bi_2 and Mg_3Sb_2 . The results, shown in Fig. R2, indicate that both materials exhibit decent tensile property. A brief discussion has also been included in the revised text.

Fig. R2 Tensile strain-stress curves of polycrystalline Mg₃Sb₂ and Mg₃Bi₂.

Comment: 2. In Figure 1e, the two blue lines lack specific labelling.

Response: Thank you for your careful reading. We have added the appropriate labels to improve readability and clarity.

Comment: 3. In the discussion of the high thermoelectric performance of Mg₃Sb_{2-x}Bi_x, although the Seebeck coefficient, electrical conductivity, and thermal conductivity were tested, there was no further analysis from a perspective on the reasons for changes in material performance parameters. It is recommended that analysis of the electronic structure and carrier mobility be included, which would help in understanding the reasons behind the improvements in thermoelectric performance more comprehensively.

Response: Thank you for your valuable suggestions. We have carefully analyzed the TE performance of Mg₃Sb_{2-x}Bi_x and added related discussions in the relevant text: "...As shown in Supplementary Fig. S4, Bi alloying has significant impacts on electrical transport properties of Mg₃Sb_{2-x}Bi_x. The downward shift of the peak *S* in Mg₃Sb_{2-x}Bi_x with higher Bi contents suggests that the bandgap is reduced, in consistency with previous reports^{30-32,34}. Additionally, when Bi content (*x*) increases to 1.5, there is an obvious rise in room temperature σ , which suggests that the grain boundary scattering is weakened due to the larger grain sizes^{30,34} (fracture morphologies

of $\text{Mg}_3\text{Sb}_{0.5}\text{Bi}_{1.5}$ in Supplementary Fig. S3). Moreover, Bi alloying leads to lower κ due to the enhanced point defect scattering of phonons...”. The improvement of TE performance in $\text{Mg}_3\text{Sb}_{2-x}\text{Bi}_x$ with Bi alloying is in consistency with previous literature (Refs. 30-32, 34), and this work emphasizes the simultaneous achievement of high TE performance and room temperature plasticity in $\text{Mg}_3\text{Sb}_{2-x}\text{Bi}_x$.

Comment: 4. Figure S13 demonstrates that the cut small blocks exhibit areas of damage. However, it is unclear whether the author placed these cut blocks within a grid for analysis, as the information in this figure is unclear and not intuitive. Additionally, it is recommended that the quality of images in Figures 3 and S12 be improved. Although Figure S12 includes a scale, it would be beneficial to additionally note the dimensions of the cut blocks for comparison. Furthermore, it would be helpful to clarify whether the small blocks shown at the bottom of Figure 3c are of the same dimensions as those in the images above.

Response: Thank you for your comments. We apologize for the unclear information about the cut blocks. In fact, the cut blocks are not placed on the grid. Instead, they remain attached to the original ingot. Because the dicing test in this work does not cut the ingot thoroughly to fully separate each other. To address this, we have added more details about dicing in the Methods section: “...The dicing did not cut the ingot thoroughly, which allowed the cut blocks to remain attached to the original ingot. Two types of cutting programs have been used. The first is a regular cut, where the ingot is cut into blocks of $150 \times 150 \mu\text{m}^2$, $100 \times 100 \mu\text{m}^2$ or $50 \times 50 \mu\text{m}^2$. The second is a gradual cut, where the cut distance decreases in every two cut steps: 200 μm , 150 μm , 100 μm , 50 μm , 20 μm , and finally 10 μm ...”. We appreciate your suggestions for improving the figures. We have enhanced their quality and added some annotations.

Comment: 5. In the discussion of the high internal resistance of the flexible TE module of $\text{Mg}_3\text{Sb}_{2-x}\text{Bi}_x$, although high internal resistance is attributed to significant interfacial resistance between $\text{Mg}_3\text{Sb}_{2-x}\text{Bi}_x$ and the electrodes, the actual test results were not provided. It is therefore necessary to include interfacial resistance tests between

Mg₃Sb_{2-x}Bi_x and the electrodes.

Response: Thank you for your suggestions. We conducted the test to measure the contact resistance between Mg₃Sb_{0.5}Bi_{1.498}Te_{0.002} and the electrodes and found that the contact resistance is very high, of about 3500 μΩ·cm², as shown in Fig. R3, which is the main obstacle preventing our flexible modules from achieving the desired output performance. We anticipate further improvement in the output performance of the module after reducing the contact resistance. We have also updated the relevant paragraphs: “...As shown in Supplementary Fig. S15, the contact resistance between Mg₃Sb_{0.5}Bi_{1.498}Te_{0.002} and Cu electrode is extremely high, of about 3500 μΩ·cm²...”.

Fig. R3 Contact resistance between Mg₃Sb_{0.5}Bi_{1.498}Te_{0.002} and Cu electrode.

Comment: 6. Similarly, when discussing the suboptimal out-of-plane performance of the thermoelectric module, which is primarily attributed to the magnitude of the temperature differential, this component should also utilize thermocouples for precise temperature measurement and analysis in order to identify the variables influencing temperature.

Response: Thank you very much for your suggestions. We indeed used thermocouples to measure the temperature difference ΔT_{module} in our flexible modules. In the out-of-plane flexible module, the measured ΔT_{module} is about 13.1 K. However, the measured

ΔT_{module} is not the actual temperature difference ΔT_{leg} exactly along the TE legs due to the existence of PI substrates, Ag paste and the high contact thermal resistance. It is quite challenging to directly measure the ΔT_{leg} of the assembled modules with the thermocouples. Instead, we calculated ΔT_{leg} based on TE parameters of the materials, as used in a previous literature (ref. 3). To avoid confusion, we have added more details in the Methods section: “ ΔT_{module} was obtained by the differences of the temperatures measured by the two thermocouples, while ΔT_{leg} was calculated by using the output voltage of the modules and the Seebeck coefficient of the TE material³”.

Reviewer #3 (Remarks to the Author):

Comment: Flexible thermoelectrics could be a potential sustainable power supply for flexible electronics. In this work, Li et al. report a high-performance Mg-based plastic semiconductor. By revealing the intrinsic plasticity in Mg₃Sb₂ and Mg₃Bi₂, and through adjusting the Sb/Bi ratios in Mg₃Sb_{2-x}Bi_x semiconductors, they attained a high zT value of ~0.72 at room temperature, as well as a large compressive strain of 43%, in polycrystalline Mg₃Sb_{0.5}Bi_{1.498}Te_{0.002}. Based on this high-performance plastic thermoelectric semiconductor, they fabricated both prototype in-plane and out-of-plane flexible thermoelectric modules, demonstrating the potential of this material for flexible electronics. In my opinion, this work is systematic, interesting, and cutting-edge in the field. Previously, flexible thermoelectrics were generally developed based on organic semiconductors or inorganic/organic hybrids with inferior thermoelectric performance. The finding of room-temperature plasticity in low-cost Mg₃Sb_{2-x}Bi_x inorganic semiconductors with good thermoelectric performance and the fabricated flexible modules will advance the future development of flexible thermoelectric technology using high-performance inorganic semiconductors. I would like to recommend this work for publication in Nature Communications after some minor revisions.

Response: Thank you very much for your careful reading, succinct overview of our work and constructive comments and suggestions.

Comment: 1. The single-crystalline Mg_3Bi_2 displays good deformability when subjected to bending and twisting in Fig. 1c. How about the single-crystalline Mg_3Sb_2 ?

Response: Thank you for your question. We have also prepared single-crystalline Mg_3Sb_2 , but it cannot be bent and twisted like single-crystalline Mg_3Bi_2 . We have added some sentences in relevant text: "...while single-crystalline Mg_3Sb_2 cannot be bent and twisted like single-crystalline Mg_3Bi_2 ...". In fact, Mg_3Sb_2 is more rigid than Mg_3Bi_2 , as evidenced in Fig. 1a, where Mg_3Sb_2 has a much higher Young's modulus. Additionally, our calculations in Fig. 2b indicate that the GSFE of Mg_3Sb_2 is higher than that of Mg_3Bi_2 , which could explain why single-crystalline Mg_3Sb_2 is less deformable than Mg_3Bi_2 .

Comment: 2. The flexible thermoelectric modules are assembled by using thin legs that were cut from the ingots. Compared to the "bottom-up" fabrication technology, i.e. ink direct-writing and sputtering, what are the advantages of the fabrication method in this work?

Response: Thank you for your insightful question. In this work, we used a conventional fabrication method for TE modules, where TE legs are obtained by cutting from the SPSeD bulk ingot. In contrast to other "bottom-up" techniques, like ink direct-writing, which requires the organic solvents to prepare the ink, the conventional cutting-assembling method could ensure high TE performance of TE legs, which is a critical factor for achieving high-performance TE modules.

Comment: 3. The authors stated the inferior output performance of the flexible module is a result of the high internal resistance. What are the reasons for this high resistance? Are there some ways to further reduce the internal resistance?

Response: Thank you for your questions. The high resistance primarily stems from high contact resistance between $\text{Mg}_3\text{Sb}_{2-x}\text{Bi}_x$ and electrodes, as shown in Fig. R3. In our view, this high contact resistance may be due to the active element Mg, particularly given the large quantity of Mg used to prepare the materials. Mg can easily react with

O₂ and H₂O in the air, forming MgO and Mg(OH)₂, which are highly resistant to electricity, thereby bringing high contact resistance between materials and electrodes. Therefore, optimizing the interface is very important for improve the output performance of the flexible module. Interface materials are commonly used to mitigate contact resistance problems. Therefore, reducing the contact resistance can be expected if optimal interface materials are used.

Comment: 4. Besides the flexible power generator, could the plastic Mg₃Sb_{2-x}Bi_x semiconductors be used for other functions in flexible electronics?

Response: Thanks for your insightful questions. Thermoelectric technology can mutually convert heat and electricity, which means it can be used not only as a power generator but also as a cooler or sensor. Given that Mg₃Sb_{2-x}Bi_x is a TE material with both high zT value and plasticity, its potential application as flexible sensors and flexible coolers for heat management/dissipation in flexible electronics can also be expected.

Comment: 5. The scale bar in Fig. 1c is missing.

Response: Thank you very much for your careful reading. We have revised Fig. 1c and added a scale bar accordingly.

Comment: 6. Fig. 3c, since different processes could result in varied mechanical properties, the methods for synthesizing Ag₂(Te,S), (Bi,Sb)₂Te₃, and Bi₂(Te,Se)₃ should be mentioned.

Response: Thank you for your constructive comments. You are correct that the fabrication process plays a crucial role in determining the mechanical properties of materials. In this work, we synthesized Ag₂(Te,S) using a melting method, consistent with our previous work. Regarding (Bi,Sb)₂Te₃ and Bi₂(Te,Se)₃, these were purchased from the company and were produced by hot-extrusion. We have provided additional details on the fabrication process in the Methods section: "...The Ag₂(Te,S) used for dicing tests were obtained by melting method. High-purity Ag shots (99.999%), Te

shots (99.999%), and S flakes (99.999%) were used and weighted accordingly, which were then loaded and sealed in the quartz tube for the melting at 1273 K. Detailed synthesis conditions can be found in our previous study¹⁹, and the commercial $\text{Bi}_2(\text{Te,Se})_3$ and $(\text{Bi,Sb})_2\text{Te}_3$ were produced by hot extrusion”.

REVIEWERS' COMMENTS

Reviewer #1 (Remarks to the Author):

All of my comments were well-addressed, I believe the article can be published in present form without any further changes.

Reviewer #2 (Remarks to the Author):

The authors reply to the comments point-by-point, and all the replies are reasonable. Moreover, the corresponding sections have been revised in the manuscript. So, I think that the manuscript can be accepted for publication.

Reviewer #3 (Remarks to the Author):

This paper has been well revised and could be accepted now.

Dear Reviewers,

Thank you very much for your positive response to our manuscript. We greatly appreciate the reviewers' valuable and positive comments.

Sincerely yours,

Tiejun Zhu

Reviewer #1 (Remarks to the Author):

Comment: All of my comments were well-addressed, I believe the article can be published in present form without any further changes.

Response: Thank you for the positive comment.

Reviewer #2 (Remarks to the Author):

Comment: The authors reply to the comments point-by-point, and all the replies are reasonable. Moreover, the corresponding sections have been revised in the manuscript. So, I think that the manuscript can be accepted for publication.

Response: Thank you for the positive comment.

Reviewer #3 (Remarks to the Author):

Comment: This paper has been well revised and could be accepted now.

Response: Thank you for the positive comment.